# Brain-wide screen of prelimbic cortex inputs reveals a functional shift during early fear memory consolidation

**Lucie Dixsaut, Johannes Gräff\***

Laboratory of Neuroepigenetics, Brain Mind Institute, School of Life Sciences, Ecole Polytechnique Fédérale de Lausanne, Lausanne, Switzerland

**Abstract** Memory formation and storage rely on multiple interconnected brain areas, the contribution of which varies during memory consolidation. The medial prefrontal cortex, in particular the prelimbic cortex (PL), was traditionally found to be involved in remote memory storage, but recent evidence points toward its implication in early consolidation as well. Nevertheless, the inputs to the PL governing these dynamics remain unknown. Here, we first performed a brain-wide, rabies-based retrograde tracing screen of PL engram cells activated during contextual fear memory formation in male mice to identify relevant PL input regions. Next, we assessed the specific activity pattern of these inputs across different phases of memory consolidation, from fear memory encoding to recent and remote memory recall. Using projection-specific chemogenetic inhibition, we then tested their functional role in memory consolidation, which revealed a hitherto unknown contribution of claustrum to PL inputs at encoding, and of insular cortex to PL inputs at recent memory recall. Both of these inputs further impacted how PL engram cells were reactivated at memory recall, testifying to their relevance for establishing a memory trace in the PL. Collectively, these data identify a spatio-temporal shift in PL inputs important for early memory consolidation, and thereby help to refine the working model of memory formation.

**\*For correspondence:**
johannes.graeff@epfl.ch

**Competing interest:** The authors declare that no competing interests exist.

## Editor's evaluation

In this study, the authors used state-of-the-art methods to perform a brain-wide screening of engram cells in the prelimbic cortex. They identified specific activity patterns of these inputs across different phases of fear memory consolidation and describe the contribution of the claustrum and insula to prelimbic inputs during encoding and recall of recent memory, respectively. This study will be of broad interest for neurobiologists studying memory.

## Introduction

The brain's ability to form enduring memories is essential for an individual's survival. Memories first need to be encoded and subsequently stored in the brain, a process that is termed memory consolidation. Memory consolidation occurs both at the scale of individual cells (cellular consolidation), which happens in the order of seconds to hours, and at the scale of brain networks (systems consolidation), which takes place in the days to weeks after learning (*Dudai, 2004*; *Dudai, 2012*). Systems consolidation across several brain areas is thought to be essential for the establishment of enduring memories (*Nadel and Hardt, 2011*).

Traditionally, the hippocampus (HPC) was demonstrated to be necessary during the early stages of memory formation and the retrieval of recent memories (which in the mouse are typically studied one day after encoding), while the medial prefrontal cortex (mPFC) was found to be rather responsible for

the consolidation and retrieval of remote memories (which are studied at least 14 days after encoding) (*Albo and Gräff, 2018*; *Frankland and Bontempi, 2005*). Indeed, multiple studies using immediate early gene mapping (IEGs, that are expressed specifically upon neuronal activation), whole brain region inhibition, or cell type-specific optogenetic manipulations (*Aceti et al., 2015*; *Frankland et al., 2004*; *Frankland et al., 2006*; *Goshen et al., 2011*; *Makino et al., 2019*; *Silva et al., 2019*; *Wheeler et al., 2013*) showed that the mPFC was predominantly important at later times as opposed to the HPC. However, recent evidence has challenged this view by highlighting a role for the mPFC also during fear memory encoding (*Bero et al., 2014*; *Cho et al., 2017*; *Cummings and Clem, 2020*; *Tang et al., 2005*; *Zelikowsky et al., 2013*) as well as for fear memory recall at recent times (*Do-Monte et al., 2015*; *Rajasethupathy et al., 2015*). The rich connectivity of the mPFC indeed places it as a potential hub region for memory consolidation, as it receives not only inputs from other cortical areas, including sensory ones, but also from various subcortical areas such as the hippocampal formation, amygdala, and thalamus (*Dixsaut and Gräff, 2021*; *Le Merre et al., 2021*), which are all implicated in memory formation (*Cho et al., 2017*; *Nonaka et al., 2014*; *Ramirez et al., 2013*; *Reijmers et al., 2007*; *Taylor et al., 2021*).

At the cellular level, mounting evidence suggests that memories are encoded and stored in *engram* cells, which, by definition (*Tonegawa et al., 2015*), are cells that are activated during the initial learning, undergo molecular and/or cellular modifications following learning, and the reactivation of which correlates with and can trigger memory recall. Engram cells have been discovered not only in the HPC (*Josselyn et al., 2015*; *Liu et al., 2012*; *Ramirez et al., 2013*), but more recently also in the mPFC (*DeNardo et al., 2019*; *Kitamura et al., 2017*; *Matos et al., 2019*). Interestingly, engram cells in the mPFC were reported to have the particular feature of staying silent until the memory is fully consolidated, although they are formed during the original learning phase (*DeNardo et al., 2019*; *Kitamura et al., 2017*; *Matos et al., 2019*). This implies that mPFC engram cells are first active during encoding, stay silent during a recent recall, and are reactivated at remote recall, although they are functionally able to trigger memory retrieval at any time.

Based on these grounds, we hypothesized that the functional contribution of mPFC inputs may change over the course of memory consolidation to govern how the mPFC engram is formed and subsequently reactivated. For this reason, we first sought out to establish a comprehensive functional map of mPFC inputs across time during fear memory consolidation, and second to analyze the downstream effect of these inputs on memory retention and mPFC engram reactivation.

## Results

### The prelimbic cortex is specifically active during the encoding of a fear memory

The mPFC is composed of the three following major areas: The anterior cingulate (ACC), the prelimbic (PL) and the infralimbic (IL) cortices (*Carlén, 2017*; *Le Merre et al., 2021*). In order to evaluate the relative activity of these subregions during the different phases of fear memory consolidation, we used a contextual fear conditioning (CFC) paradigm in combination with cFos immunohistochemistry (IHC), an IEG marker of neuronal activity (*Pérez-Cadahía et al., 2011*). We measured the freezing percentage of wild-type (WT) mice at CFC encoding before the footshocks occurred, as well as at recent recall 1 day post-encoding and at remote recall 14 days post-encoding. Each group was controlled for by a no shock group. We observed that both at recent and remote recall, mice display a significantly higher freezing percentage than the no shock control groups (*Figure 1B*), indicating successful memory formation.

We then quantified cFos expression in the three subregions of the mPFC (*Figure 1C*) 90 min after the corresponding behavioral session. We found that while all regions were more active at encoding than during the memory recalls, only in the PL did we observe a significant increase in cFos compared to the no shock control groups (*Figure 1C–F*). These results indicate that the mPFC as a whole, and the PL in particular, are activated by the encoding of a contextual fear memory. In turn, this finding suggests an important role of the PL already during this early phase of memory consolidation, which is coherent with the formation of engram cells in the PL at the time of encoding (*DeNardo et al., 2019*; *Kitamura et al., 2017*; *Matos et al., 2019*).

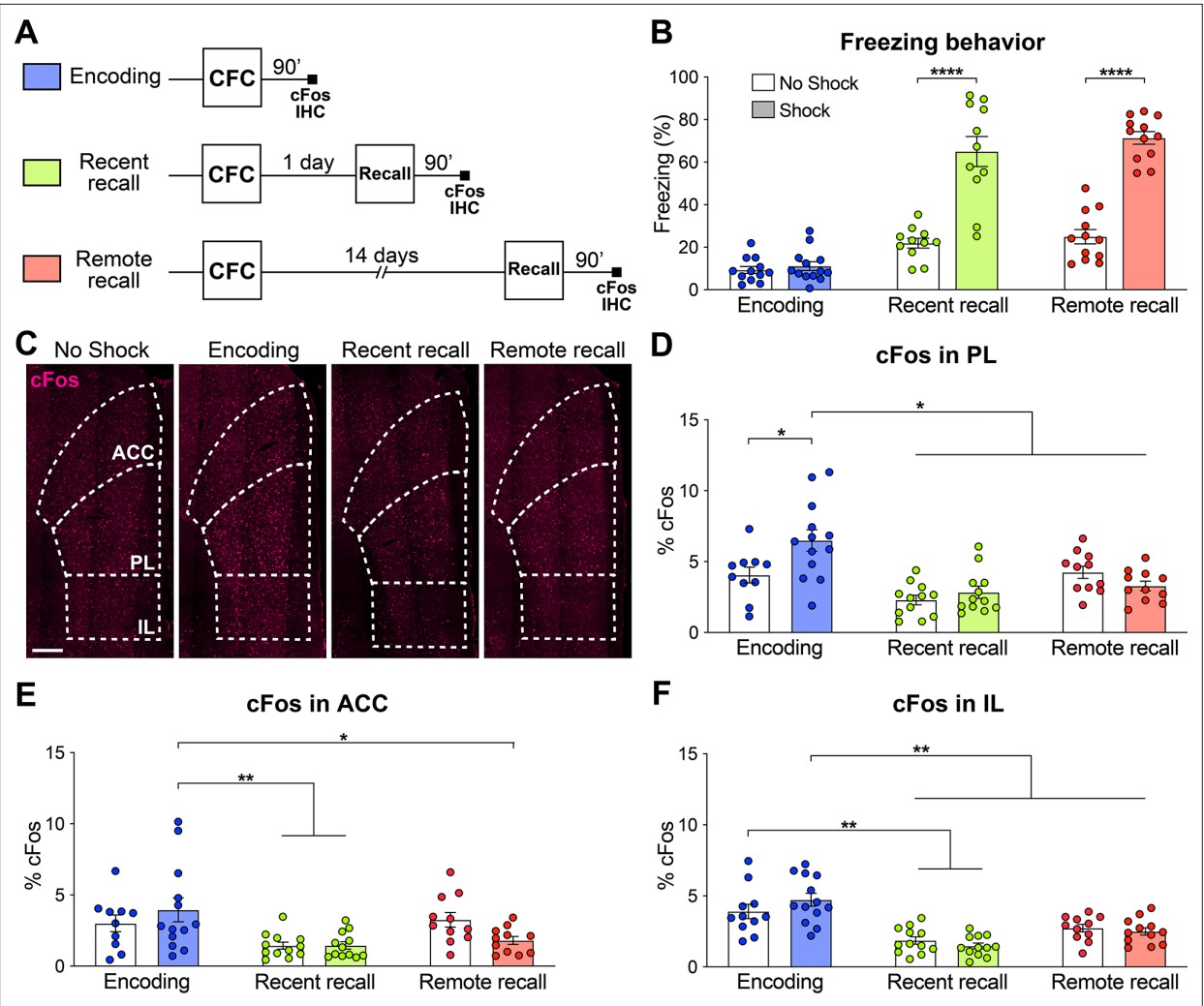

**Figure 1.** The prelimbic cortex (PL) is activated by the encoding of a contextual fear memory. (**A**) Experimental design. For encoding, mice were perfused 90 min after contextual fear conditioning (CFC). For recent and remote recalls, mice were perfused 90 min after a 1 day and a 14 day recall, respectively. (**B**) Percentage freezing measured during the 3 min of habituation before the shocks (encoding, in blue), at recent (in green) or remote (in red) memory recalls, for the animals undergoing CFC (shock, filled) and the control groups that were exposed to the CFC chamber without the shock (no shock, clear). Two-tailed unpaired t-tests, ****: p<0.0001. At recent recall, Cohen's d=2.48; at remote recall, Cohen's d=4.24. (**C**) Representative images of cFos immunostainings in the medial prefrontal cortex (mPFC). Scale: 250 µm. (**D–F**) Percentage of cFos over DAPI in (**D**) PL (one-way-ANOVA, $F_{(5, 63)}$=9.172, p<0.0001), (**E**) anterior cingulate cortex (ACC) (one-way ANOVA, $F_{(5, 63)}$=4.394, p=0.0017) and (**F**) infralimbic cortex (IL) (one-way ANOVA, $F_{(5, 65)}$=13.34, p<0.0001). (**D–F**) Stars represent least significant p-values of Tukey's multiple comparisons tests: *: 0.01 < p < 0.05; **: 0.001 < p < 0.01. n=11–13 animals per group.

The online version of this article includes the following source data for figure 1:

**Source data 1.** Raw data for *Figure 1*.

## Brain-wide screen of PL engram inputs

Next, we aimed to identify PL inputs that could be responsible for this peaked PL activity at encoding and for the establishment of its engram during memory consolidation (*DeNardo et al., 2019*; *Kitamura et al., 2017*; *Matos et al., 2019*). To this end, we employed an activity-dependent monosynaptic retrograde tracing technique (*Figure 2A and B*). Specifically, we used the TRAP2 mouse line (*DeNardo et al., 2019*), in which the *Fos* promoter drives the expression of the tamoxifen-dependent Cre[ERT2] recombinase. These mice were first injected in the PL with helper adeno-associated viruses (AAVs) expressing Cre-dependent nuclear GFP, the TVA receptor, and the optimized rabies glycoprotein oG. Thereby, the expression of these proteins in PL engram cells could be triggered with tamoxifen injection at the time of encoding. Three weeks post-encoding, we injected a modified

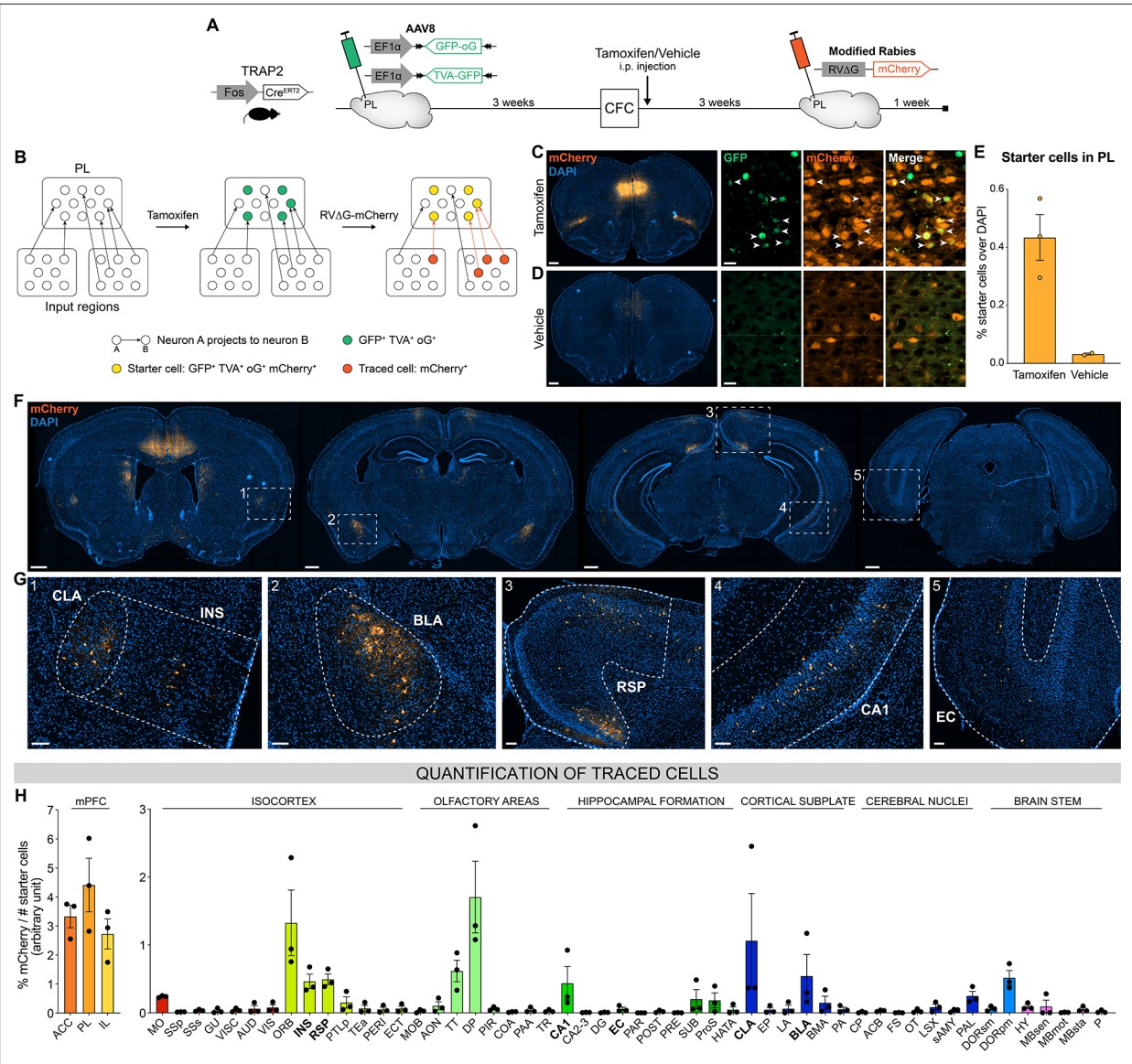

**Figure 2.** Brain-wide retrograde tracing identifies monosynaptic inputs of prelimbic cortex (PL) engram cells. (**A–B**) Experimental design and timeline. *Fos-Cre^ERT2* animals were first injected in the PL with helper adeno-associated viruses (AAVs) expressing GFP, TVA receptor, and oG (rabies optimized glycoprotein) in a Cre-dependent manner. Tamoxifen (or vehicle for control) was injected right after contextual fear conditioning (CFC) to trigger recombination in cFos+ cells. Three weeks later, a modified rabies virus (RVΔG-mCherry with EnvA coating) was injected in PL where it infected TVA-expressing cells, replicated in oG-expressing cells, and was retrogradely transsynaptically transported. A week later, brains were collected to quantify monosynaptic inputs of PL engram cells labelled with mCherry. (**C, D**) Representative images of the PL injection site (scale 400 μm) and magnified view of starter cells (scale 20 μm) with tamoxifen (**C**) or vehicle (**D**) injection. (**E**) Percentage of starter cells over DAPI in PL. (**F**) Representative images of traced cells throughout the brain (scale 500 μm). (**G**) Magnified views of traced cells (scale 100 μm) in claustrum (CLA) (inset 1), insular cortex (INS) (1), basolateral amygdala (BLA) (2), retrosplenial cortex (RSP) (3), ventral CA1 (vCA1) (4), and entorhinal cortex (EC) (5). (**H**) Brain-wide quantification of traced cells, normalized by the number of starter cells for each animal, in the medial prefrontal cortex (mPFC) subregions (left) and the rest of the brain (right). Tamoxifen: n=3 animals; vehicle: n=2 animals.

The online version of this article includes the following source data and figure supplement(s) for figure 2:

**Source data 1.** Raw data for *Figure 2* and its supplement.

**Source data 2.** Abbreviations for the brain regions.

**Figure supplement 1.** Raw quantifications of the rabies tracing experiment.

rabies vector RVΔG-mCherry (*Wickersham et al., 2007*) that can only infect and replicate in TVA- and oG-expressing cells, respectively, allowing us to transsynaptically label all monosynaptic inputs of PL engram cells (*Figure 2B*). As expected, we found that tamoxifen injection increased the number of starter cells expressing both GFP and mCherry in PL compared to vehicle (*Figure 2C–E*), confirming the specificity of these tools to restrict tracing to PL engram cells.

We then quantified the percentage of traced cells (mCherry⁺) throughout the brain (*Figure 2F–H*). Although most traced cells were found in the mPFC itself and its neighboring areas (orbitofrontal cortex, OFC, and dorsal peduncular area, DP, *Figure 2H*), we observed traced inputs in several other brain regions, notably the claustrum (CLA, *Figure 2G* inset 1), insular cortex (INS, inset 1), basolateral amygdala (BLA, inset 2), retrosplenial cortex (RSP, inset 3), CA1 field of the HPC (mostly the ventral part, inset 4), taenia tecta (TT), thalamus polymodal association cortex-related areas (DORpm), subiculum (SUB), and to a lesser extent the entorhinal cortex (EC, inset 5). Without tamoxifen injection, traced cells were negligible (*Figure 2—figure supplement 1*), which further confirms the rabies tracing specificity.

With this approach, we identified relevant PL inputs that might be responsible for the development of the PL engram cell population, but we still lacked information on whether, when, and the extent to which these inputs are activated across memory consolidation.

## PL inputs are differentially activated across memory consolidation

Out of the regions projecting to PL engrams, we selected six brain areas with consistent input tracing for further investigation, because of their previously documented implication in various aspects of fear memory: The EC, for its role in memory formation (*Roy et al., 2017*) and its known projection to the mPFC necessary at encoding (*Kitamura et al., 2017*) and retrieval (*Pilkiw et al., 2022*); the RSP for its necessity for recent (*Cowansage et al., 2014*) and remote fear memory recall (*Todd et al., 2016*); the INS for its requirement during the consolidation and expression of contextual fear memories (*Alves et al., 2013*), as well as for its regulation of fear expression (*Gehrlach et al., 2019*; *Klein et al., 2021*); vCA1 for its importance for CFC encoding (*Kim and Cho, 2020*) and recent recall (*Jimenez et al., 2020*); the BLA for the role of BLA to PL projections in memory encoding (*Kitamura et al., 2017*; *Klavir et al., 2017*) and PL to BLA projections in memory recall (*Do-Monte et al., 2015*; *Kitamura et al., 2017*); and the CLA as CLA to EC projections are necessary during memory encoding (*Kitanishi and Matsuo, 2017*) and for its involvement in attention (*Atlan et al., 2018*).

To assess the relative activity of these PL inputs during fear memory consolidation, we needed a tracing technique that could be coupled with neuronal activity measurements from encoding to remote recall, which cannot be achieved with rabies tracing from engram cells. Therefore, we combined conventional retrograde tracing with neuronal activity-dependent cFos staining: Injection of AAVretro-GFP in PL prior to any behavioral test allowed to trace all anatomical projections to the PL (*Figure 3A and B*), while cFos IHC 90 min after CFC encoding, recent and remote memory recall allowed to assess the activation of these projections (*Figure 3A*). In each region we measured cFos as well as GFP-traced inputs (*Figure 3—figure supplements 1 and 2*), thus controlling for homogenous tracing across behavioral groups. Next, we compared the pattern of activation between PL projectors only and the region as a whole to highlight the specific recruitment of PL projectors, and we focused on the associative information conveyed in this activity by normalizing it to the no shock control groups (*Figure 3*, see *Figure 3—figure supplements 1 and 2* for quantifications only normalized to chance level).

We first investigated cortical areas projecting to the PL: EC (specifically layer 5, comprising most of EC traced cells, *Figure 3C*), RSPag (which contained most of RSP traced cells, *Figure 3G*), and INS (*Figure 3K*). In the EC, we observed a significant activation of PL projections at encoding compared to both recent and remote recalls (*Figure 3E*), which was not the case in total cFos quantifications (*Figure 3F*). This suggests a specific recruitment of EC neurons projecting to PL (EC → PL) at encoding. In contrast, RSPag and INS displayed a different pattern of activation, as there was no activation in RSPag → PL and INS → PL projections at encoding, but during recent memory recall (*Figure 3I and M*, respectively). Compared to total cFos in both regions, this activity was again specific to PL projectors (*Figure 3J and N*, respectively).

Second, we investigated PL inputs in subcortical areas: vCA1 (*Figure 3O*), BLA (*Figure 3S*), and CLA (*Figure 3W*). In vCA1, we observed no differential recruitment of vCA1 → PL projections

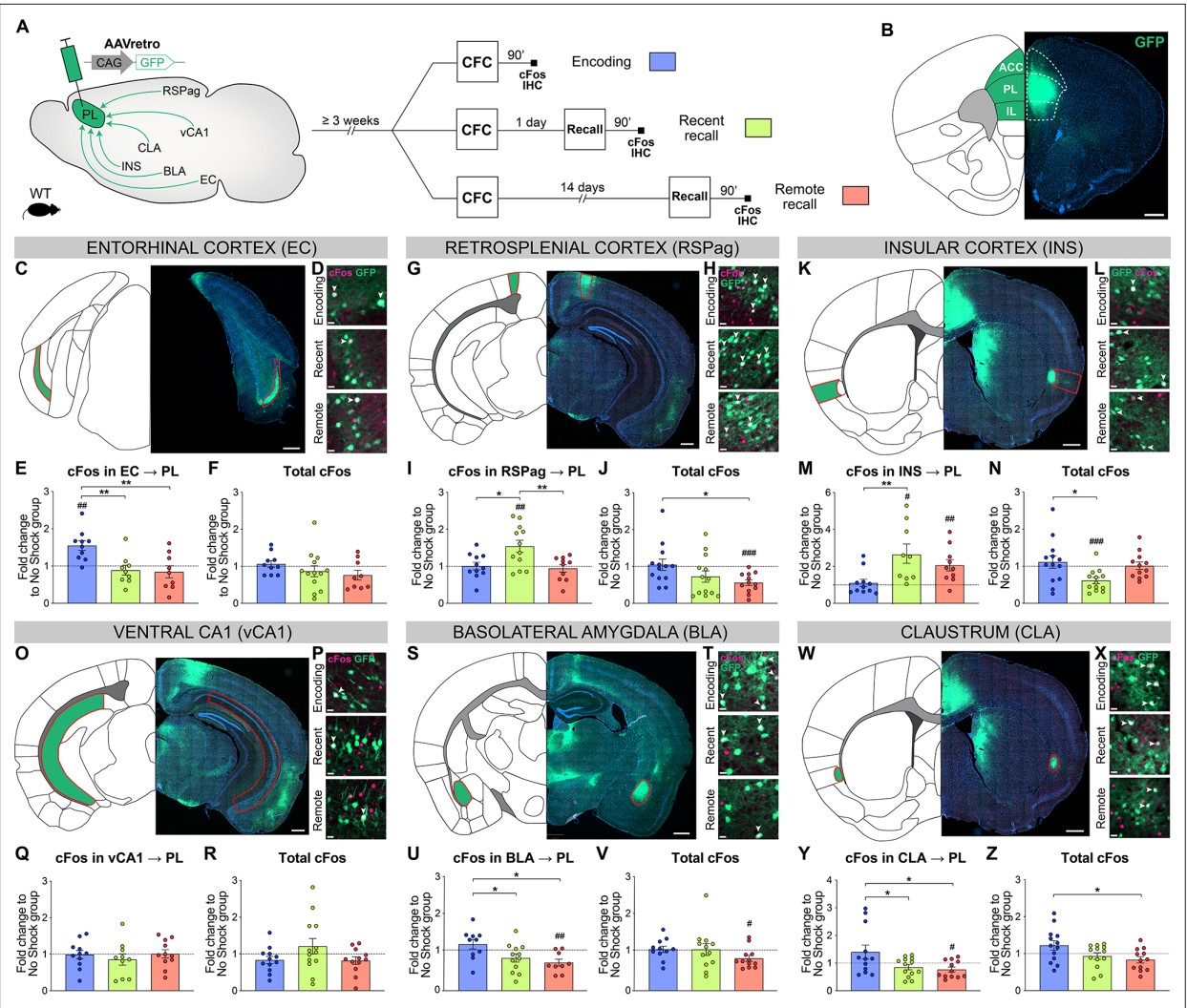

**Figure 3.** Prelimbic cortex (PL) inputs are differentially activated during memory consolidation. (**A**) Experimental design, injection of AAVretro-GFP in the PL for input tracing, and quantification of activation by cFos immunostaining 3 weeks later at either contextual fear conditioning (CFC) encoding (blue), recent (green), or remote (red) recall. Brains were collected 90 min after the behavior session. (**B**) Representative image of AAVretro-GFP injection site in the PL region of the medial prefrontal cortex (mPFC). Scale: 500 μm. (**C–Z**) For each region: Representative image of PL input tracing, magnified view of GFP and cFos at encoding, recent, and remote time points (all scales: 20 μm); quantifications of cFos in PL projections and total cFos in the input region, expressed as fold change to the no shock control group. Note that cFos in PL projections values were first normalized by chance level for each animal (see *Figure 3—figure supplement 1*). (**C–F**) EC (**C**, scale 500 μm), (**E**) cFos in EC → PL (one-way ANOVA, $F_{(2,25)}$=8.153, p=0.0019) and (**F**) total cFos. (**G–J**) RSPag (**G**, scale 400 μm), (**I**) cFos in RSPag → PL (one-way ANOVA, $F_{(2,35)}$=3.275, p=0.0497) and (**J**) total cFos (one-way ANOVA, $F_{(2,35)}$=3.275, p=0.0497). (**K–N**) INS (**K**, scale 500 μm), (**M**) cFos in INS → PL (one-way ANOVA, $F_{(2,27)}$=5.405 p=0.0106) and (**N**) total cFos (one-way ANOVA, $F_{(2,35)}$=4.583, p=0.0171). (**O–R**) vCA1 (**O**, scale 400 μm), (**Q**) total cFos and (**R**) cFos in vCA1 → PL. (**S–V**) basolateral amygdala (BLA) (**S**, scale 500 μm), (**U**) cFos in BLA → PL (one-way ANOVA, $F_{(2,28)}$=4.922, p=0.0147) and (**V**) total cFos. (**W–Z**) claustrum (CLA) (**W**, scale 500 μm), (**Y**) cFos in CLA → PL (one-way ANOVA, $F_{(2,34)}$=4.502, p=0.0184), and (**Z**) total cFos (one-way ANOVA, $F_{(2,35)}$=3.833, p=0.0313). Stars represent p-values of Tukey's multiple comparisons tests (*: 0.01 < p < 0.05; **: 0.001 < p < 0.01), hashtag signs represent p-values of two-tailed one-sample t-tests comparing the difference to 1, which represents levels of the no shock controls (#: p≤0.05; ##: 0.001 < p ≤ 0.01; ###: p≤0.001). n=9–13 animals per group.

The online version of this article includes the following source data and figure supplement(s) for figure 3:

**Source data 1.** Raw data for *Figure 3* and its supplements.

**Figure supplement 1.** Complementary quantifications for the activation of prelimbic cortex (PL) inputs in entorhinal cortex (EC), RSPag and insular cortex (INS).

**Figure supplement 2.** Complementary quantifications for the activation of prelimbic cortex (PL) inputs in vCA1, basolateral amygdala (BLA), and claustrum (CLA).

between different times of memory consolidation (*Figure 3Q, R*; *Figure 3—figure supplement 2A*). However, the elevated cFos expression in the vCA1 as a whole at encoding as well as in the no shock group supports the role of HPC in context exploration (*Figure 3—figure supplement 2B*; *Schiller et al., 2015*). For the BLA, we found a significant activation of BLA → PL projections at encoding compared to the recalls (*Figure 3U*), which is not the case for total cFos in BLA (*Figure 3V*). The recruitment of BLA → PL projection at encoding is in agreement with its importance during fear memory formation (*Kitamura et al., 2017*; *Klavir et al., 2017*). Interestingly, we observed the same pattern of activation in CLA → PL projections (*Figure 3Y*), together with an overall higher activation of the whole CLA region at encoding compared to remote recall (*Figure 3Z*).

Taken together, we found that PL inputs from the EC, BLA, and CLA were active only at encoding, while RSPag and INS projections were recruited during recent memory recall.

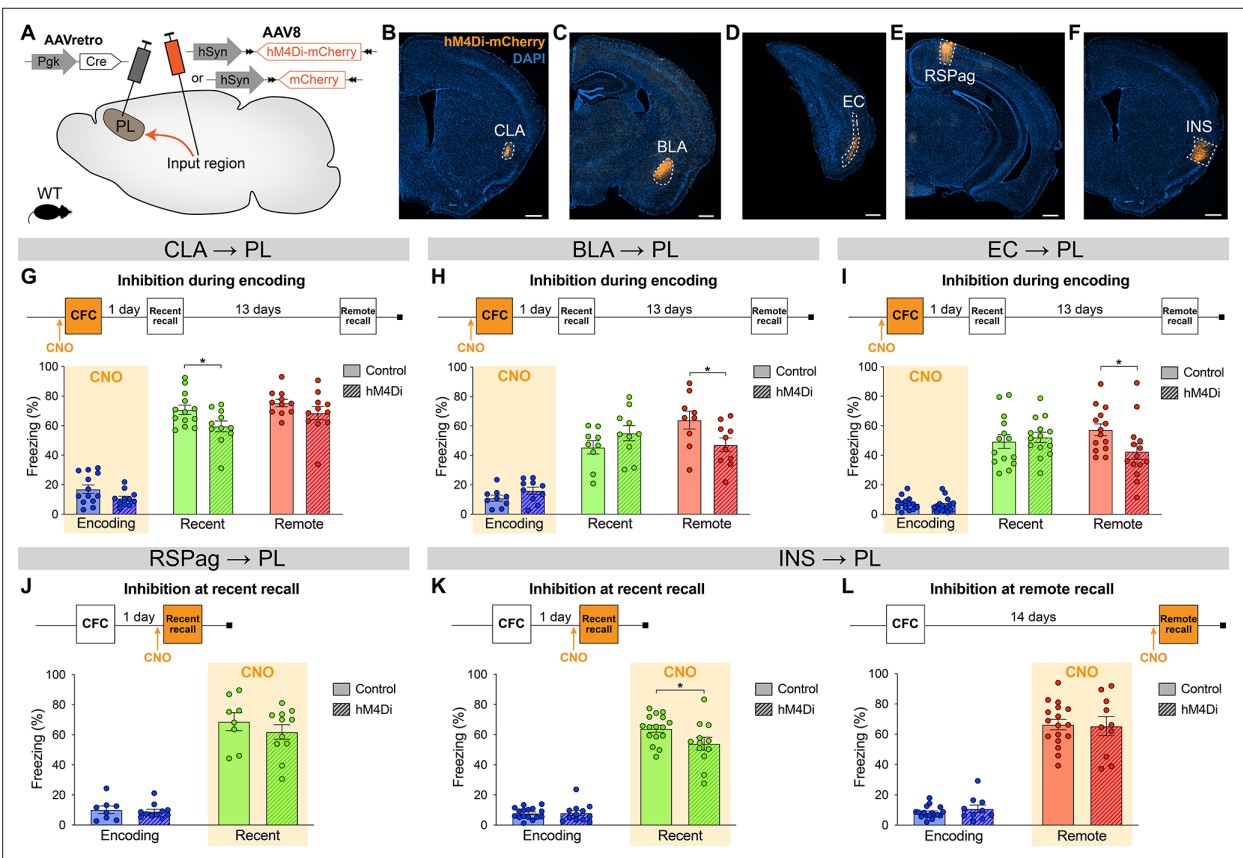

**Figure 4.** Chemogenetic manipulation of prelimbic cortex (PL) inputs reveals the functional importance of claustrum (CLA) projections at encoding and insular cortex (INS) projections at recent memory recall. (**A**) Experimental design. AAVretro-Cre was injected in the PL, and AAV-DIO-hM4Di-mCherry (or AAV-DIO-mCherry for controls) in the desired input region in order to specifically inhibit the projections to the PL upon clozapine-*N*-oxide (CNO) injection. Representative images of the injection site in the input region for CLA (**B**), basolateral amygdala (BLA) (**C**), entorhinal cortex (EC) (**D**), RSPag (**E**), and insular cortex (INS) (**F**), all scales 500 µm. Experimental timeline and freezing percentage of (**G**) CLA → PL inhibition during encoding (at recent recall, Cohen's d=−0.95), (**H**) BLA → PL inhibition during encoding (at remote recall, Cohen's d=−1.02), (**I**) EC → PL inhibition during encoding (at remote recall, Cohen's d=−0.84), (**J**) RSPag → PL inhibition during recent recall, (**K–L**) INS → PL inhibition during recent (Cohen's d=−0.83) (**K**) and remote (**L**) recall. Stars represent p-values of two-tailed unpaired t-tests between hM4Di and control groups (*: p≤0.05). n=8–17 animals per group.

The online version of this article includes the following source data and figure supplement(s) for figure 4:

**Source data 1.** Raw data for *Figure 4* and its supplements.

**Figure supplement 1.** Quantification of the chemogenetic inactivation of claustrum (CLA) → prelimbic cortex (PL) projections.

**Figure supplement 2.** Claustrum (CLA) → prelimbic cortex (PL) inhibition after encoding does not impair memory recall and does not alter locomotion and exploration behavior.

**Figure supplement 3.** Inhibition of RSPag → prelimbic cortex (PL) at recent recall does not impair reconsolidation at later times.

## PL inputs are functionally relevant at different stages of memory consolidation

Next, in order to establish whether the differential activity in PL inputs across memory consolidation is also functionally relevant, we selectively inhibited each projection at the time(s) when they were most active and tested subsequent memory retention. We used the designer receptor exclusively activated by designer drug (DREADD) receptor hM4Di, which upon clozapine-*N*-oxide (CNO, the DREADD agonist) administration inhibits neuronal activity (*Roth, 2016*). We targeted hM4Di expression to specific PL inputs by injecting AAVretro-Cre into the PL, and AAV-DIO-hM4Di-mCherry (or AAV-DIO-mCherry for controls) in the desired input region (*Figure 4A–F*).

First, we assessed the functionality of projections that were active at encoding, namely the CLA, BLA, and EC. For the CLA → PL inhibition, we observed an impaired memory at recent recall (*Figure 4G*). To confirm that CNO indeed inhibits hM4Di-expressing neurons, we expressed it in CLA → PL neurons (*Figure 4—figure supplement 1A*) and perfused the animals 90 min after CFC to stain for cFos (*Figure 4—figure supplement 1B, C*). While the percentages of hM4Di+ and cFos+ cells were equivalent in both groups (*Figure 4—figure supplement 1D, E*), the amount of double positive hM4Di+cFos+ cells was significantly decreased with CNO injection, confirming the inhibition of projection neurons during behavior (*Figure 4—figure supplement 1F*). Furthermore, this behavioral result was not due to a delayed effect of CNO injection, as inhibiting CLA → PL projections right after encoding did not result in impaired memory at any time point (*Figure 4—figure supplement 2A,B*). Likewise, the effect of CLA → PL inhibition was not due to an unspecific effect on locomotion and exploratory behavior as tested in an open field arena (*Figure 4—figure supplement 2C-E*). In contrast to the effect of CLA → PL inhibition, when BLA → PL and EC → PL projections were inhibited during encoding, we observed an impairment of remote memory recall for both, while recent recall was unaffected (*Figure 4H, I*, respectively). These results indicate that while the BLA → PL and EC → PL projections are important at encoding for the consolidation of remote memories, as shown previously (*Kitamura et al., 2017*), the CLA → PL projection is important at encoding for recalling recent memories.

Next, we tested the functionality of projections that were most active during fear memory recall, namely the RSPag and INS to PL projections. We found that although the RSPag → PL projection was specifically active at recent recall (*Figure 3I*), its inhibition during this time did not affect memory retrieval (*Figure 4J*). It is also unlikely that the RSPag → PL projection is involved in the expression or reconsolidation of fear memories later on, as memory performance was unchanged at later times after recent recall inhibition (*Figure 4—figure supplement 3A,B*). However, it is still possible that this projection could play a role in extinction of fear memories, which would need to be tested in future experiments. Of further note, it was recently reported that although the entire RSP is necessary for recent and remote recall, it is rather the granular subregion of the RSP and not the RSPag that is responsible for this effect, suggesting a dissociated role of the two RSP subregions which could explain our observations (*Tsai et al., 2022*).

Conversely, INS → PL projection inhibition during recent recall resulted in decreased freezing (*Figure 4K*), positing this pathway to be important for recent fear memory expression. In contrast, consistent with no significant activation of the INS → PL projection at remote memory recall, the inhibition of this projection did not result in any behavioral effect (*Figure 4L*).

Taken together, these findings indicate that CLA, BLA, and EC projections to the PL are required at encoding for proper memory formation, but with different time implications. While the BLA and EC connections are important for recalling remote memories, the CLA projection is specifically important for recalling recent ones. In addition, recent memory recall is also under the influence of the INS → PL projection, since its inhibition at this time led to significant memory impairment. This suggests a progressive functional shift in PL projections regulating memory consolidation.

## PL engram reactivation correlates with memory retrieval when CLA or INS inputs are inhibited

Lastly, we decided to further investigate the effect of CLA and INS input inhibition on engram reactivation in the PL. We hypothesized that if the inhibition of a specific PL input results in memory impairment, then the reactivation of the original PL engram, established at the time of memory encoding, may also be altered. Indeed, engram reactivation has been correlated with memory retention in BLA

(*Reijmers et al., 2007*), and artificial engram reactivation in the PL (*Kitamura et al., 2017*) or HPC (*Liu et al., 2012*) has been found to trigger memory recall. In order to measure engram reactivation, we used the *Fos::tTA* mouse line (*Reijmers et al., 2007*), expressing the doxycycline (Dox)-dependent tTA transcription factor under the *Fos* promoter, which we injected with AAV-TRE-GFP into the PL 3 weeks before CFC (*Figure 5A*). As tTA specifically binds the TRE (tetracycline responsive element) promoter in the absence of Dox, this approach allows for inducible long-term expression of GFP in PL engram cells during a desired time-window (*Figure 5B, C*). In combination with chemogenetic inhibition of projection neurons as previously described (*Figure 4*), we then silenced selective PL inputs and assessed the degree of engram reactivation between CFC encoding and recall, by measuring cFos and GFP overlap in the PL (*Figure 5C*).

First, we focused on the CLA → PL inhibition at memory encoding (*Figure 5D*). Behaviorally, this approach confirmed the impaired memory at recent memory recall as observed in WT mice (*Figure 5E*, see also *Figure 4G*). Furthermore, we found a significant correlation between PL engram cells reactivation, measured as double positive GFP$^+$cFos$^+$ cells normalized to the total number of GFP$^+$ cells, and freezing at recent recall, which was observed only in the CNO group (*Figure 5F*). No differences were observed in overall GFP$^+$, cFos$^+$, double positive GFP$^+$cFos$^+$, and total reactivation percentages in PL between CNO- and vehicle-treated animals. (*Figure 5—figure supplement 1A–D*). In contrast, when we inhibited the CLA → PL projection during encoding and tested remote recall, memory was not impaired (*Figure 5—figure supplement 2B*, see also *Figure 4G*) and we observed no effect on PL engram reactivation (*Figure 5—figure supplement 2C–G*). Importantly, this correlation was not a side effect of CNO injection as we observed no behavioral nor engram reactivation differences in control virus-injected animals receiving vehicle or CNO (*Figure 5—figure supplement 3A–C*). This result indicates that CLA → PL inhibition during encoding modifies PL engram reactivation at recent recall only, and that following this inhibition, the less animals reactivate the original PL engram, the less they recall the fear memory.

Second, we inhibited the INS → PL projection at recent recall, which also confirmed the behavioral effect on memory retrieval (*Figure 5H*) in WT mice (*Figure 4K*). Similar to the CLA results, there was no difference between the CNO and vehicle groups in the percentage of GFP$^+$, cFos$^+$, double positive GFP$^+$/cFos$^+$ cells, or total reactivation (*Figure 5—figure supplement 1E–H*). However, we again observed a significant correlation in the CNO group between PL engram reactivation and freezing at recent recall (*Figure 5I*), indicating that INS → PL inhibition at recent recall impairs recent memory retrieval and modifies PL engram reactivation.

These findings suggest that the spatiotemporal shift in the activity and functionality of PL projections during memory consolidation also occurs at the level of PL engram cells.

## Discussion

In this study, we investigated the role of specific PL inputs during the course of fear memory consolidation. Using an unbiased tracing approach combined with pathway-specific chemogenetic inhibition experiments, we discovered a novel functional implication of two PL inputs, namely from the CLA and INS, and confirmed the role of two others, from the BLA and EC. More precisely, we found that the CLA → PL projection is important at encoding for contextual fear memories, specifically for recent memory recall, while the INS → PL pathway is required for memory expression during recent recall. Furthermore, our findings confirm previous reports that the BLA → PL and EC → PL projections are functionally relevant at encoding for remote memory recall (*Kitamura et al., 2017*).

These results expand the existing literature on memory consolidation and refine the working model of memory formation and retrieval. Importantly, our data add to the growing evidence on the importance of the mPFC during early phases of memory consolidation (*Bero et al., 2014*; *Cho et al., 2017*; *Rajasethupathy et al., 2015*; *Takehara-Nishiuchi et al., 2020*; *Zelikowsky et al., 2013*). Thereby they further challenge the standard theory of memory consolidation, which posits that the HPC is necessary for encoding and recent recall, while the mPFC would take over from the HPC only at remote recall (*Frankland and Bontempi, 2005*). Indeed, we observe a significant activation of the PL during memory encoding already, as well as the functional implication of several of its inputs at encoding and recent recall, which advances the temporal engagement of the mPFC to earlier than remote recall only.

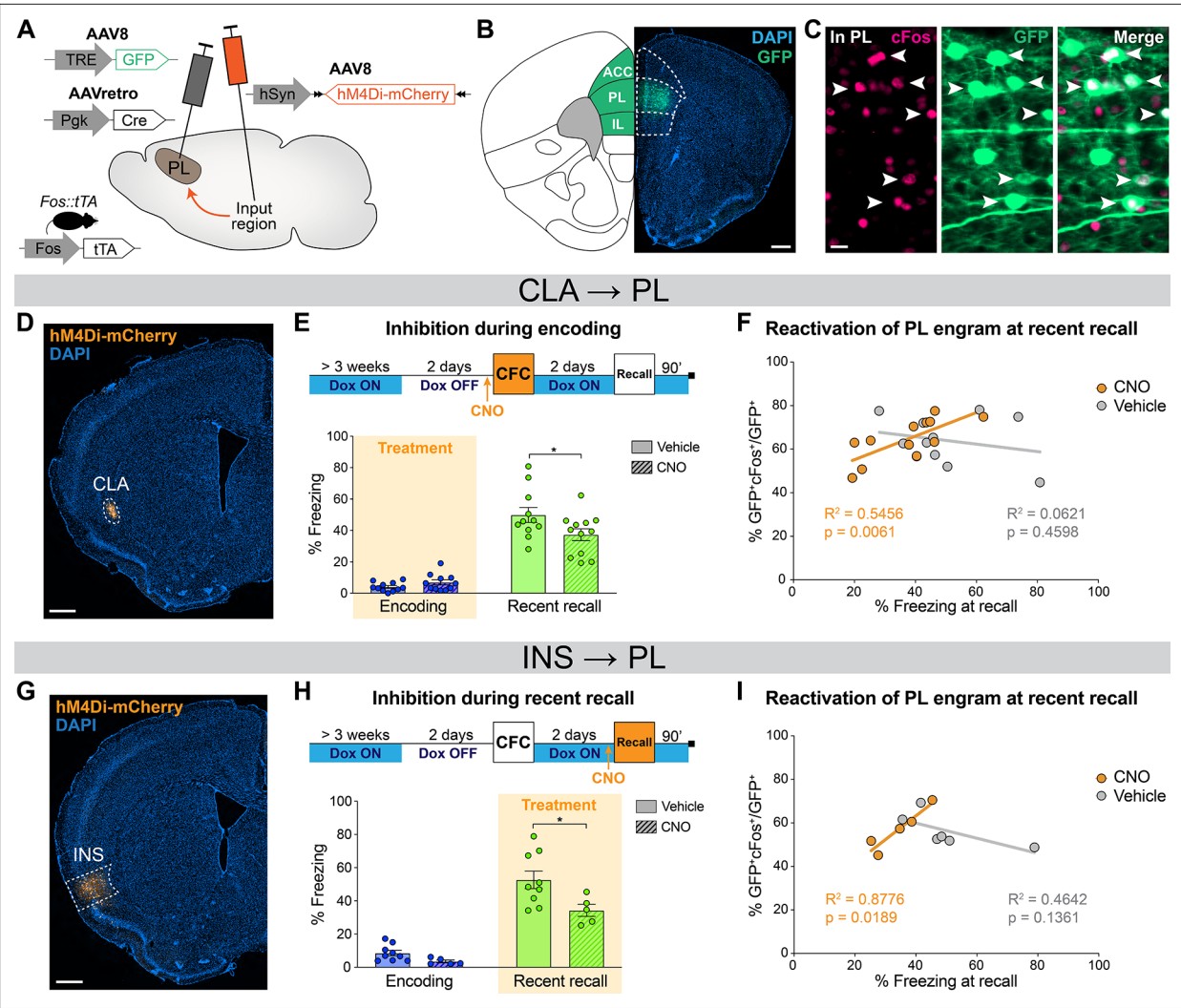

**Figure 5.** Prelimbic cortex (PL) engram reactivation correlates with freezing when claustrum (CLA) or insular cortex (INS) inputs are inhibited. (**A**) Experimental design. 3 weeks before behavior started, *Fos::tTA* mice were injected with AAVretro-Cre in PL and AAV-DIO-hM4Di-mCherry in the desired input region, as well as AAV-TRE-GFP in the PL, so that GFP was only expressed in cFos$^+$ cells in the absence of doxycycline (Dox). (**B**) GFP expression at the PL injection site (scale 400 μm). (**C**) Magnified view in the PL (scale 20 μm) of reactivated engram cells, indicated by white arrows. (**D**) Representative image of the CLA input region. (**E**) Experimental timeline (top) and freezing percentage (bottom) during recent memory recall when CLA → PL projections were inhibited during encoding (Cohen's d=–0.87). (**F**) Reactivation of PL engram cells (%GFP$^+$cFos$^+$/GFP$^+$) at recent recall for CLA → PL inhibition, correlated with freezing percentage at recent recall for clozapine-*N*-oxide (CNO) (orange) and vehicle (gray) groups. (**G**) Representative image of the INS input region. (**H**) Experimental timeline (top) and freezing percentage (bottom) during recent memory recall when INS → PL projections were inhibited during recent recall (Cohen's d=–1.33). (**I**) Reactivation of PL engram cells (%GFP$^+$cFos$^+$/GFP$^+$) at recent recall for INS → PL inhibition, correlated with freezing percentage at recent recall for CNO (orange) and vehicle (gray) groups. (**E, H**) Stars represent p-values of two-tailed unpaired t-tests between CNO and vehicle groups (*: p≤0.05). (**F, I**) Correlations assessed with linear regressions, R$^2$ and p-values are reported on the graphs. n=11–12 (CLA) or 5–9 (INS) per group.

The online version of this article includes the following source data and figure supplement(s) for figure 5:

**Source data 1.** Raw data for *Figure 5* and its supplements.

**Figure supplement 1.** Complementary quantifications for the engram reactivation analysis in the prelimbic cortex (PL).

**Figure supplement 2.** Claustrum (CLA) → prelimbic cortex (PL) inhibition during encoding does not affect remote recall and remote engram reactivation in PL.

**Figure supplement 3.** Clozapine-*N*-oxide (CNO) administration per se does not alter memory recall or engram reactivation.

At remote memory recall, in contrast, we did not observe any PL input that was engaged or behaviorally relevant. To our knowledge, such an input has never been reported, although the importance of the mPFC as a whole (**Bero et al., 2014**; **Frankland et al., 2004**; **Goshen et al., 2011**), and of its functional outputs at remote recall is well established (**Do-Monte et al., 2015**; **Kitamura et al., 2017**). This apparent gap, or the impossibility to trace back the flow of information upstream of the mPFC, could be explained if its inputs are distributed across a vast network after memory consolidation. In that case, they could potentially be redundant and therefore harder to functionally identify. As such, these results are in agreement with the multiple trace theory (**Moscovitch et al., 2006**; **Nadel and Moscovitch, 1997**), which posits that the HPC first encodes memory upon learning, but does not store the memory per se. Rather, it is a distributed network of cortical neurons – with inputs from the HPC – that contains pieces for long-term information storage. With time, this leads to the creation of multiple traces in the brain for a given memory, making it more stable and less likely to be disrupted, as we have observed here.

Corroborating the implication of the PL during the early phases of memory consolidation, we found that the CLA → PL is required at encoding, which is the first report that this PL input is functionally important during fear memory formation. The CLA has known roles in attention (**Atlan et al., 2018**) and context exploration (**Kitanishi and Matsuo, 2017**), which are likely to support its role in memory formation. Interestingly, we observed that the CLA → PL projection was important at encoding only for recent, and not for a later remote recall. This result suggests that this projection has a time-limited effect on memory consolidation, and that other brain areas allow for a proper remote recall and thereby compensate in case of the CLA → PL inhibition at encoding. Of note, a CLA → EC projection has also been reported to be necessary at CFC encoding for recent recall (**Kitanishi and Matsuo, 2017**), while here we have observed that the EC → PL projection is functional at encoding for remote recall. This opens the possibility of an indirect circuit from the CLA to the PL via the EC, which could compensate in case the CLA → PL direct connection is impaired, but this remains to be experimentally determined. Nevertheless, these results imply a shift in PL circuits underlying the encoding of contextual fear memory, reminiscent of findings that reported a shift in PL circuits when an auditory fear memory is retrieved (**Do-Monte et al., 2015**).

In line with the notion of an overall shift in PL circuits during early memory consolidation, we report that the INS → PL projection is relevant at recent, but not at remote memory recall. The INS as a whole has classically been involved in taste learning (**Yiannakas and Rosenblum, 2017**) and the encoding of conditioned taste aversion (**Sano et al., 2014**), but has also been implicated in recent CFC recall (**Alves et al., 2013**) as well as in auditory fear memory extinction (**Klein et al., 2021**). The INS to mPFC reciprocal connectivity has only been investigated in the context of taste learning, where it was recently found necessary for the expression of novel taste aversion (**Kayyal et al., 2021**). However, a direct role of INS input to PL during fear memory consolidation has not been described before. This finding therefore supports a broader role for the INS in learning beyond taste-related tasks (**Boughter and Fletcher, 2021**; **Shi et al., 2020**).

Since the CLA and INS manipulations both resulted in impaired recent memory recall, we decided to assess PL engram cells for their reactivation, a characteristic of engram cells that is linked with memory performance during recall (**Kitamura et al., 2017**; **Liu et al., 2012**; **Reijmers et al., 2007**). We found that the reactivation of PL engram cells significantly correlated with freezing behavior, but only when the CLA and INS inputs were inhibited, and not in controls. This finding raises the question of the functionality of PL engram cells at recent recall in normal conditions. Recently, a concept of 'silent' engram cells in the PL has been developed, which postulates that silent PL engram cells have the particular feature of not being activated by recent recall – although their artificial reactivation can trigger recall at this time – but of becoming active only at remote recall (**Kitamura et al., 2017**; **Matos et al., 2019**). Our findings are thus in line with this concept: When the CLA → PL projection is inhibited at encoding, or when the INS → PL projection is inhibited at recent recall, overall engram reactivation is left unchanged, but recent memory expression is impaired, which would suggest that the PL engram population itself is not required for recent memory recall. Since upon inhibition we observe the emergence of a correlation between PL engram cells reactivation and memory retention, it is possible that CLA and INS inputs target inhibitory neurons in the PL, which normally prevent engram reactivation during recent recall, thus allowing the PL engram cells to stay functionally silent. Releasing this inhibition could perturb normal memory expression, explaining the observed memory

impairment at recent recall. Indeed, it has been reported that CLA → mPFC targets inhibitory neurons (*Jackson et al., 2018*), and that PL interneurons are necessary for memory encoding (*Cummings and Clem, 2020*), but this hypothesis remains to be tested at the level of PL engram cells.

Alternatively, we can hypothesize that the activation of these projections, instead of their inhibition, would also disrupt the constraint on functional engram reactivation by increasing local inhibition of principal neurons, which could lead to memory impairment in an equivalent manner. Overall, the magnitude of the memory impairment that we have observed and the fact that projection-specific manipulation did not impair overall engram reactivation suggest that other projections and/or regions are contributing in addition to the ones we manipulated. The extensive connectivity between regions, and notably from the CLA, could therefore explain the apparent resilience of engrams to small perturbations in the memory network.

There are several experimental limitations that accompany these findings. First, as we only used male mice in this study, these results cannot be generalized across sexes. Second, the use of two different engram-tagging mouse lines, TRAP2 for the rabies tracing and *Fos::tTA* for the engram reactivation experiments, was dictated by technical constraints. A TRE-dependent rabies tracing system was not readily available at the start of this study, and the use of a Cre-dependent system for chemogenetic inhibition precluded the use of the TRAP2 line again for the engram reactivation experiments. However, as these two mouse lines are both *Fos* promoter-based (*DeNardo et al., 2019*; *Reijmers et al., 2007*), we would not expect major differences with the use of one or the other lines. Indeed, rabies tracing from PL engram cells using the *Fos::tTA* line has been published since, and the reported input areas are all also found in our brain-wide screen (*Kitamura et al., 2017*). Third, the use of two different retrograde tracing viruses raises the question of tropism: RVΔG and AAVretro have been reported to not trace the exact same set of input regions, notably with AAVretro being biased toward cortical inputs (*Sun et al., 2019*). We could have therefore missed some regions due to preferential input tracing. Another limitation is the relatively slow kinetics of chemogenetic inhibition. As CNO is injected 30 min before behavior, we cannot exclude that compensation mechanisms may take over, especially in the case of remote recall inhibition, which would prevent a functional isolation of the targeted projection during behavior as reported previously (*Goshen et al., 2011*). By restricting the inhibition to the smallest possible period, the use of optogenetics could allow to visualize the consequences of this inhibition in real time in future experiments. Lastly, differences in timing and strength of behavioral protocols could explain discrepancies with other studies. For example, using cFos IHC, we did not observe an increased activity at remote recall in mPFC regions as opposed to previous findings (*Frankland et al., 2004*). Unified conditioning protocols could help to clarify these.

These limitations notwithstanding, here we have shown that PL circuits undergo a spatiotemporal shift during contextual fear memory consolidation, with claustral inputs being critical at encoding, and insular cortical inputs at recent memory recall. Our results therefore support a dynamic and distributed nature of memory formation and storage.

## Materials and methods

### Animals

All animals and procedures used in this study were approved by the Veterinary Office of the Federal Council of Switzerland under the animal experimentation licenses VD2808.1 and VD2808.2. C57Bl/6JR WT male mice were purchased from Janvier Labs (France) around 6–7 weeks of age and left for at least 1 week before the beginning of the experiments. *Fos::Cre^{ERT2}* (TRAP2) animals were bred in house from the original JAX strain #030323 on a C57Bl/6JR background. *Fos::tTA* male mice were bred in house from the original JAX strain #018306 on a C57Bl/6JR background. Animals were housed in a 12 hr light/dark cycle with water and food available ad libitum. All animals were group-housed except for the input tracing experiment where they were single caged 2 days before the end of the experiment. They were all handled by the experimenter for at least 3 days before the first behavioral procedure to reduce stress levels.

All behavioral procedures were performed between 1 pm and 5 pm local time and animals were randomly assigned to experimental groups.

## Viral stereotaxic injections

### Procedure

At 7–8 weeks, animals were anesthetized with a mix of fentanyl (0.05 mg/kg), midazolam (5 mg/kg), and metedomidin (0.5 mg/kg), i.p. After shaving and subcutaneous injection of a local anesthetic mix (lidocaine 6 mg/kg and bupivacaine 2.5 mg/kg), the animal was placed on a stereotaxic frame. The skin was disinfected with betadine and opened with a scalpel. The skull was thoroughly cleaned, the orientation of the head was adjusted, and holes were drilled at the desired coordinates with a 0.5 mm drill bit. The virus was loaded into pulled glass capillaries (intraMARK, Blaubrand, tip diameter 10–20 µm), and injected to the target area at a speed of 100 nL/min. The needle was left in place for 5 min, and slowly pulled up to limit backflow. After all injections were done, the skin was sutured (5/0 Prolene, Ethicon), the animal was injected i.p. with atipamezol (2.5 mg/kg) to reverse the anesthesia, and placed back in a heated cage. After surgery, the animals were administered paracetamol in the drinking water for a week (Dafalgan, 1 mg/mL).

### Viruses

The following viruses were used in this study:

- AAV-DIO-TVA-2A-oG (Salk Institute Vector Core, serotype 8), here referred to as AAV-DIO-TVA-oG. Titer: $8.78 \times 10^{12}$ GC/mL, mixed 1:1 with AAV-FLEX-GFP-oG (see below).
- AAV-EF1a-FLEX-H2B-GFP-P2A-oG (Salk Institute Vector Core, serotype 8), here referred to as AAV-FLEX-GFP-oG. Titer: $3.93 \times 10^{12}$ GC/mL, mixed 1:1 with AAV-DIO-TVA-oG; total injection volume in PL: 400 nL.
- Modified rabies virus RVΔG-mCherry, EnvA pseudotyped (Salk Institute Vector Core, SADB19 strain). Titer: $3.5 \times 10^8$ ifu/mL. Injection volume in PL: 300 nL.
- AAV-CAG-GFP (Addgene 37825, retrograde serotype), here referred to as AAVretro-GFP. Titer: $7 \times 10^{12}$ GC/mL; injection volume in PL: 200 nL.
- AAV-pgk-Cre (Addgene 24593, retrograde serotype), here referred to as AAVretro-Cre. Titer: $1.7 \times 10^{13}$ GC/mL; injection volume in PL: 200 nL.
- AAV-hSyn-DIO-hM4D(Gi)-mCherry (Addgene 44362 or Zürich VVF v84, serotype 8), here referred to as AAV-DIO-hM4Di-mCherry. Titer: $1.8 \times 10^{13}$ GC/mL (Addgene – diluted ½) or $4.5 \times 10^{12}$ GC/mL (VVF); injection volume: 150–200 nL depending on the regions.
- AAV-hSyn-DIO-mCherry (Addgene 50459, serotype 8), here referred to as AAV-DIO-mCherry. Titer: $2.3 \times 10^{13}$ GC/mL, diluted ½; injection volume: 150–200 nL depending on the regions.
- AAV-TRE3G-GFP (UNC Vector Core, serotype 8), here referred to as AAV-TRE-GFP. Titer: $4.1 \times 10^{12}$ GC/mL, mixed 1:1 with AAVretro-Cre; injection volume in PL: 250 nL.

### Injection coordinates from bregma

- PL: AP +2.0; ML ±0.35; DV -2.2.
- EC: AP -4.15; ML ±3.55; DV -4.3.
- RSPag: AP –2.6; ML ±1.1; DV –0.6.
- INS: AP +1.0; ML ±3.85; DV –4.0 in WT mice, or AP +1.0; ML ±3.9; DV –4.0 in *Fos::tTA* mice.
- BLA: AP -1.0; ML ±3.15; DV -4.55.
- CLA: AP +1.0; ML ±3.2; DV –4.0 in WT mice, or AP +1.0; ML ±3.1; DV –4.0 in *Fos::tTA* mice.

For input tracing experiment with AAVretro-GFP, animals were injected unilaterally. For all other experiments, animals were injected bilaterally.

## Behavioral procedures

### Contextual fear conditioning

CFC encoding and recall were performed in the same chamber (TSE Systems). CFC encoding consisted in a first 3 min exploration phase, followed by three 2 s long 0.8 mA footshocks spaced by 28 s. After the last shock, the animal was left in the chamber for an additional 15 s and brought back to its home cage. The recall consisted in a 3 min exposure to the same context, without any shock. For all experiments except the engram reactivation, recent recall took place 1 day after the encoding and remote recall 14 days later. For the engram reactivation experiment, recent recall took place 2 days after the encoding, to leave enough time for GFP expression. The movement of the animals was automatically measured using an infrared beam cut detection system (TSE Systems). Freezing detection threshold

was set at 1 s of immobility. No shock control animals underwent the same procedure but did not receive any shocks. Animals without any chemogenetic manipulation were excluded if they froze less than 20% of the time during the recall (in total two animals were excluded in all experiments).

### Tamoxifen injection

In the rabies tracing experiment, TRAP2 mice were injected i.p. with tamoxifen (4-hydroxytamoxifen, Sigma-Aldrich, CAS 68392-35-8, 50 mg/kg) immediately after CFC. Tamoxifen was prepared as follows: powdered tamoxifen was dissolved in ethanol 100% at a concentration of 20 mg/mL and stored at –20°C. On the day of the experiment, tamoxifen was re-dissolved by shaking at 37°C, 2 volumes of corn oil were added and ethanol was evaporated shaking at 37°C, for a final concentration at 10 mg/mL. Tamoxifen was kept at 37°C until injection to prevent precipitation.

### Chemogenetic inhibition

In these experiments, mice were injected i.p. with CNO (Sigma-Aldrich, CAS 34233-69-7, 3 mg/kg) 30 min before the desired behavioral phase. CNO was prepared as follows: 5 mg of CNO were resuspended in 50 µL of DMSO and stored at –20°C. On the day of the experiment, CNO was diluted 1/500 in NaCl to reach a concentration of 0.2 mg/mL, and injected at the desired volume. In the engram reactivation experiment, control animals were injected with an equivalent volume of vehicle i.p. (0.9% NaCl, B. Braun, and 1/500 DMSO).

### Open field test

For CNO control experiments, 30 min after CNO injection the animals were placed in a large circular arena and left to freely explore for 15 min. Video-tracking of the animals and locomotion quantification was automatically performed using the EthoVision software (Noldus).

### Engram reactivation

*Fos::tTA* mice were administered Dox (Sigma-Aldrich, CAS 24390-14-5) in the drinking water at 0.2 mg/mL. Dox was prepared as follows: Powdered Dox was resuspended in water from the animal facility at 50 mg/mL, aliquoted and frozen at –20°C until further use. It was then diluted in water bottles to reach a concentration of 0.2 mg/mL. Dox was administered at least 3 weeks before the behavioral protocol started, and was refreshed every week. In order to open the tagging-window, Dox was removed 2 days before encoding, and administered back right thereafter for the remaining time of the protocol.

### Sample size and behavioral replicates

No statistical method was used to predetermine sample size. The number of animals used in each experiment was the minimum required to obtain statistical significance, based on our experience with this behavioral paradigm and in agreement with standard literature. Data from the input tracing experiment was pooled from three independent batches (*Figures 1 and 3* and their supplements). Data from the rabies tracing experiment comes from one batch (*Figure 2* and its supplement). Data from the chemogenetic manipulation in WT was pooled from at least two independent batches for each manipulation (*Figure 4*). Data from the CNO controls comes from one batch each (*Figure 4—figure supplements 1 and 2C–F*). Data from the engram reactivation was pooled from one to two batches (*Figure 5* and its supplements). In all graphs, one dot represents one animal.

## Histology

Ninety min after the last behavioral test, animals were anesthetized with pentobarbital (150 mg/kg, Streuli Pharma) and transcardially perfused with first 1× PBS and then 4% paraformaldehyde (PFA) in 1× PBS. Brains were extracted, post-fixed overnight in 4% PFA, transferred in cryoprotectant (30% sucrose in 1× PBS) for at least 48 hr, and frozen at –80°C. Sections of 20 µm were cut using a cryostat and kept free-floating in an antifreeze solution (30% ethylene glycol, 15% sucrose, 0.02% azide in 1× PBS) until staining.

For cFos immunostaining, sections were incubated in blocking buffer (1% BSA, 0.3% Triton-X in 1× PBS) for 90 min at room temperature, followed by primary antibody incubation for two nights at 4°C in antibody dilution buffer (1% BSA, 0.1% Triton-X in 1× PBS). After four washes in 1× PBS + 0.1%

Triton-X, they were incubated with secondary antibody in antibody dilution buffer for 2 hr at room temperature, stained with Hoechst (1:10,000 in 1× PBS, Invitrogen #H3570) for 5 min and washed three times before mounting on glass slides and covered with Fluoromount-G mounting medium (Southern Biotech). Images were acquired on an Olympus slide scanner VS120 L100 with a 20× objective.

For the input tracing experiment, a primary antibody goat anti-cFos (1:1000, Santa Cruz #sc-52-G, RRID: AB_2629503) with a secondary antibody donkey anti-goat AF-647 (1:1000, Thermo Fisher Scientific #A21447, RRID: AB_2535864) was used. For all other experiments, a primary antibody rabbit anti-cFos (1:1000, Synaptic Systems #226003, RRID: AB_2231974) with a secondary antibody donkey anti-rabbit AF-647 (1:1000, Thermo Fisher Scientific #A31573, RRID: AB_2536183) was used. GFP and mCherry signals were not amplified.

For verification of the injection sites, six sections per animal were taken spanning the injection site, stained with Hoechst and mounted. Images were acquired on an Olympus slide scanner VS120 L100 with a 10× objective.

For the rabies tracing experiment, one every four sections of 20 μm were mounted on Superfrost slides (Fisher scientific) and stained with Hoechst, before imaging on an Olympus slide scanner VS120 L100 with a 10× objective.

For simplicity and clarity in the text, we used 'DAPI' to refer to nuclei stained with Hoechst.

## Image analysis

Images were analyzed using QuPath (v0.1.4 to v0.3.1) (*Bankhead et al., 2017*), by an experimenter blinded to the groups.

For the rabies tracing experiment (*Figure 2*), every section was aligned to the reference Allen Brain Atlas using a Fiji plug-in developed by the bioimaging platform at EPFL (*Chiaruttini et al., 2022*), before using a QuPath custom-built script for cell detection and classification (see supplementary material). It used multiple rounds of the built-in 'Cell Detection' plug-in (once for each channel, plus one for DAPI). After detection, cells are given a classification based on the overlap of their coordinates to the DAPI channel detections.

For the input tracing experiment (*Figure 3*), two to three sections per brain region per animal were manually annotated based on the Allen Brain Reference Atlas, and identification of the detected GFP$^+$ and cFos$^+$ cells within each annotated region was established using the custom-made QuPath script. An animal was excluded from further analysis if the percentage of traced inputs in a given area was below a region-specific threshold, as the amount of traced cells was region-dependent (thresholds were EC: 2%; RSPag: 1%; INS: 0.5%; vCA1: 0.5%; BLA: 0.6%; CLA: 1%). The chance ratio was calculated as (GFP$^+$cFos$^+$/DAPI)/chance level, where chance level was calculated as (GFP$^+$/DAPI)x(cFos$^+$/DAPI). Then, chance ratios were further normalized by the averaged chance ratio of the matching no shock control groups (*Figure 3*). cFos$^+$ cells in mPFC (*Figure 1*) were quantified in the non-injected contralateral mPFC using three to four sections per animal.

For the chemogenetic manipulation experiments (*Figures 4–5*), animals were excluded if the hM4Di-mCherry signal was leaking outside of the target region or if the amount of infected cells was too low. cFos$^+$ and hM4Di$^+$ in CLA (*Figure 4—figure supplement 1*) were quantified using the QuPath custom-built script, in three to four sections per animal.

For engram reactivation experiments (*Figure 5*), cFos$^+$ and GFP$^+$ cells were quantified in PL using the QuPath custom-built script, in three to four sections per animal. Animals were excluded from further analysis if the percentage of GFP was below 1%.

## Statistics

All statistics and graphical representations were done with GraphPad Prism 9. All data are represented in mean ± SEM, with one dot representing one animal in all graphs. Data from the input tracing screen were analyzed using ordinary one-way ANOVAs, and further comparisons were performed with Tukey's multiple comparisons tests between all groups (alpha = 0.05). In case of normalizations, difference to 1 was analyzed using two-tailed one-sample t-tests (alpha = 0.05). Data from the chemogenetic manipulation and engram reactivation experiments were analyzed using two-tailed unpaired t-tests between the two groups (alpha = 0.05), and correlations were assessed with simple linear regressions. Statistical differences in freezing percentage are all accompanied by a measure of the effect size with a calculation of Cohen's d.

## Acknowledgements

We would like to thank all past and current members of the Laboratory of Neuroepigenetics for their support and discussion throughout this project, in particular Verena Doblmayr for her contribution to the behavioral experiments and histological analysis, Liliane Glauser for overall technical assistance and mouse genotyping, and Gabriel Berdugo-Vega for insightful discussions on the project. We would like to also thank the EPFL Bioimaging platform (BIOP) for their continuous support in image acquisition and analysis, and the EPFL Center of Phenogenomics (CPG) for ensuring animal welfare at all times. This work was supported by the European Research Council (ERC-2015-StG 678832), the Swiss National Science Foundation (SNSF, 31003A_155898), the National Competence Center for Research SYNAPSY (51NF40-185897), and the Floshield and Dragon Blue Foundations. JG is a MQ fellow and NARSAD Independent Investigator.

## Additional information

### Funding

| Funder | Grant reference number | Author |
| --- | --- | --- |
| NARSAD | Independant Investigator Grant 24497 | Johannes Gräff |
| European Research Council | ERC-2015-StG 678832 | Johannes Gräff |
| Swiss National Science Foundation | 31003A_155898 | Johannes Gräff |
| National Center for Competence in Research (NCCR) "SYNAPSY" | PH02P35 | Johannes Gräff |
| MQ fellow | MQ15FIP100012 | Johannes Gräff |
| Floshield Foundation | | Johannes Gräff |
| Dragon Blue Foundation | | Johannes Gräff |

The funders had no role in study design, data collection and interpretation, or the decision to submit the work for publication.

### Author contributions

Lucie Dixsaut, Conceptualization, Data curation, Formal analysis, Investigation, Methodology, Visualization, Writing – original draft, Writing – review and editing; Johannes Gräff, Conceptualization, Funding acquisition, Project administration, Supervision, Writing – original draft, Writing – review and editing

### Author ORCIDs

Johannes Gräff http://orcid.org/0000-0002-3219-3578

### Ethics

All animals and procedures used in this study were approved by the Veterinary Office of the Federal Council of Switzerland under the animal experimentation licenses VD2808.1 and VD2808.2. Every effort was made to minimize suffering.

### Decision letter and Author response

Decision letter https://doi.org/10.7554/eLife.78542.sa1
Author response https://doi.org/10.7554/eLife.78542.sa2

## Additional files

### Supplementary files
• MDAR checklist

- Source code 1. Script for image analysis on QuPath 0.3, to detect and classify colocalized cells.

## Data availability

All data analyzed during this study are included in the source data files.

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
