## [Editor Report]

In this study, the authors used state-of-the-art methods to perform a brain-wide screening of engram cells in the prelimbic cortex. They identified specific activity patterns of these inputs across different phases of fear memory consolidation and describe the contribution of the claustrum and insula to prelimbic inputs during encoding and recall of recent memory, respectively. This study will be of broad interest for neurobiologists studying memory.

---

## [Decision Letter]

**Decision letter after peer review:**

Thank you for submitting your article "Brain-wide screen of prelimbic cortex inputs reveals a functional shift during early fear memory consolidation" for consideration by *eLife*. Your article has been reviewed by 3 peer reviewers, including Olivier J Manzoni as Reviewing Editor and Reviewer #1, and the evaluation has been overseen by Laura Colgin as the Senior Editor.

Essential revisions:

Additional experiments, less than 2 months' work:

1. While the dose of CNO used is low (3mg/kg), the control (i.e., vehicle) for the crucial Figure 5 is not adequate. CNO should be tested in mice injected with mCherry instead of AV-DIO-hM4Di-mCherry. Side effects of CNO metabolites are a growing concern, and proper monitoring would increase the confidence and significance of these results.

2. An experiment may help with understanding the apparent contrasting results in the investigation of RSP to PL projections: while RSP to PL neurons are active during recent memory recall when inactivated, no effect has been reported in memory performances. Testing memory performances later (without further manipulation), should reveal if RSP->PL neurons are involved in the following reconsolidating and extinguishing of the memory trace.

Must do, no additional experiments are required:

1. An effect size measure (such as Cohen's d) should accompany the main statistical differences.

2. A typo in Figures 3S1 and 3S2 referring to the images in Figure 2 should be Figure 3?

3. The sex of the mice should be indicated in the abstract and the generalization of the results to females should be discussed in the discussion.

4. line 155: Figure 3N should be 3Y?

5. line 156: Figure 3M should be 3Z?

6. Although mathematically correct, the correlation between %GFP+cFos+/GFP+ and freezing at recall is only a weak indicator of engram reactivation. This seems especially true for CLA->PL inhibition during encoding, which has a significant but marginal impact on freezing. This can easily make sense given the extensive connectivity of the claustrum, but it would benefit from being to be discussed.

7. Would chemogenetic activation (rather than inhibition) during encoding (CLA->PL) or recent recall (INS->PL) saturate/prevent/disrupt engram reactivation? This point should at least be discussed, considering that the "functional" findings are based solely on inhibition.

8. Please remove the claims of necessity when referring to the results of selective inactivation of PL afferents.

9. Please discuss alternative interpretations of the data in the discussion.

*Reviewer #1 (Recommendations for the authors):*

While in general, the experiments are carefully designed and the interpretations cautious, three points need clarification.

First, although the dose of CNO used is low (3mg/kg), the control (i.e., vehicle) for the crucial Figure 5 is not adequate. CNO should be tested in mice injected with mCherry instead of AV-DIO-hM4Di-mCherry. Side effects of CNO metabolites are a growing concern, and proper monitoring would increase the confidence and significance of these results.

Second, although mathematically correct, the correlation between %GFP+cFos+/GFP+ and freezing at recall is only a weak indicator of engram reactivation. This looks especially true for the inhibition of the CLA->PL during encoding which significantly but marginally impacts freezing. This may be logical considering the claustrum's widespread connectivity, but it needs to be discussed.

Third, would chemogenetic activation (rather than inhibition) during encoding (CLA->PL) or recent recall (INS->PL) saturate/occlude/perturbate the reactivation of engrams? In addition to satisfying my curiosity, one must consider that the "functional" conclusions are solely based on inhibition.

*Reviewer #2 (Recommendations for the authors):*

The conclusions of the paper are supported by data and the proper controls were conducted. I found this paper important and of potential interest in the field of memory consolidation and cortical engrams contribution. I have no major points to raise.

However, I want to suggest an experiment that may help understand the apparent contrasting results in the investigation of RSP to PL projections.

In fact, while RSP to PL neurons are active during recent memory recall when inactivated, no effect has been reported in memory performances. This could be explained by RSP to PL involvement in the following reconsolidating and extinguishing the memory trace. If this is the case, testing memory performances at a later time point (without further manipulation), should reveal this effect. Moreover, none of the previous experiments in which the authors inactivate CLA, BLA and EC has an effect during the CNO treatment but later on, during recall. With the same logic, also inhibition during recent recall may be likely to have an effect later on, and not during the treatment session itself.

*Reviewer #3 (Recommendations for the authors):*

1. Beautiful figures.

2. The inclusion of an immediate shock control is a useful non-associative control for contextual fear conditioning.

3. A measure of effect size (such as Cohen's d) should accompany key statistical differences.

4. Typo in supp Figure 3S1 and 3S2 referencing images from Figure 2 should be Figure 3?

5. The sex of the mice should be stated in the abstract and the generalizability of the results to females should be discussed in the Discussion.

---

## [Author Response]

Essential revisions:Additional experiments, less than 2 months' work:1. While the dose of CNO used is low (3mg/kg), the control (i.e., vehicle) for the crucial Figure 5 is not adequate. CNO should be tested in mice injected with mCherry instead of AV-DIO-hM4Di-mCherry. Side effects of CNO metabolites are a growing concern, and proper monitoring would increase the confidence and significance of these results.

The reviewer correctly pointed out that we cannot exclude that our observations could be side effects from CNO metabolites and not originate from the inhibition of specific projections, despite the low concentration of CNO used here (3mg/kg). Therefore, we conducted a new experiment as suggested, where all animals received a control virus in the CLA, and tested the effect of Vehicle vs CNO injection in absence of DREADD receptors (Figure 5 – new figure supplement 3).

As shown, this manipulation did not impair memory performance at recent recall, as expected (Figure 5 S3B); we also did not observe any correlation between engram reactivation (%GFP^+^cFos^+^/GFP^+^) and freezing at recent recall in both the vehicle and CNO groups (Figure 5 S3C). We can therefore conclude that CNO itself or its metabolites are not responsible for the correlation that we observed previously (Figure 5F). These results are now presented in the revised version of the manuscript.

2. An experiment may help with understanding the apparent contrasting results in the investigation of RSP to PL projections: while RSP to PL neurons are active during recent memory recall when inactivated, no effect has been reported in memory performances. Testing memory performances later (without further manipulation), should reveal if RSP->PL neurons are involved in the following reconsolidating and extinguishing of the memory trace.

We thank the reviewer for raising this important point. Indeed, from our previous experiments, we could not assess the possibility that RSPag projections to PL are involved in the reconsolidation or extinction of the memories later on. To test this, we conducted a new experiment in which the animals underwent the same procedure as before, with inactivation of the RSPag to PL projection at recent recall (which again did not impair memory recall), and tested the same animals 1 day (recall 2) and 13 days (recall 3) later, without CNO injection. The new figure 4 —figure supplement 3 presents the results of this experiment. As no memory impairment was visible in both recall 2 and recall 3, we conclude that RSPag projections to PL are unlikely to be involved in memory reconsolidation at recent recall. These results are now presented in the revised version of the manuscript.

Must do, no additional experiments are required:1. An effect size measure (such as Cohen's d) should accompany the main statistical differences.

As a measure of the effect size allows to better estimate the importance of a statistical difference, we have calculated Cohen’s d as suggested by the reviewer for all statistical differences in freezing behavior, and added it to the corresponding figure legends. We revised the methods section accordingly.

2. A typo in Figures 3S1 and 3S2 referring to the images in Figure 2 should be Figure 3?

Thank you for noticing this mistake, it has now been corrected.

3. The sex of the mice should be indicated in the abstract and the generalization of the results to females should be discussed in the discussion.

The reviewer is right in pointing out that we only used male mice in this study, therefore our results cannot be generalised to female mice unless formally tested. We added this precision in the abstract and in the discussion as suggested.

4. line 155: Figure 3N should be 3Y?5. line 156: Figure 3M should be 3Z?

Thank you for noticing these mistakes, they have now been corrected.

6. Although mathematically correct, the correlation between %GFP+cFos+/GFP+ and freezing at recall is only a weak indicator of engram reactivation. This seems especially true for CLA->PL inhibition during encoding, which has a significant but marginal impact on freezing. This can easily make sense given the extensive connectivity of the claustrum, but it would benefit from being to be discussed.

It is correct that none of our projection-specific manipulations resulted in a complete memory impairment, although the effect size of the observed differences is always larger than 0.8, and that the correlation between engram reactivation and freezing is significant but not strong. Taken together, this suggests that other regions or projections are also important in these processes and constrain the magnitude of the observed effects. As the reviewer suggested, the extensive connectivity, notably of the CLA, supports this view. We have therefore added this important consideration in the discussion.

7. Would chemogenetic activation (rather than inhibition) during encoding (CLA->PL) or recent recall (INS->PL) saturate/prevent/disrupt engram reactivation? This point should at least be discussed, considering that the "functional" findings are based solely on inhibition.

We thank the reviewer for raising this interesting point of discussion. We therefore discussed this aspect in the revised version of the manuscript. We initially hypothesized that CLA and INS projections target inhibitory interneurons in the PL which role is to constrain the functional reactivation of the engram population within a specific window. As such, inhibiting the CLA or INS projections to PL resulted in a modification of the functionality of PL engram reactivation. Alternatively, we can hypothesize that the activation of these projections, instead of their inhibition, would also disrupt functional engram reactivation constraint by increasing local inhibition of principal neurons, which could lead to memory impairment in an equivalent manner.

8. Please remove the claims of necessity when referring to the results of selective inactivation of PL afferents.

The reviewer is right in pointing out that we never obtained a complete memory impairment following the selective inhibition of PL inputs, and that we did not observe a global change in engram reactivation following these manipulations. Consequently, we have tamed the claims of “necessity” (and now refrain from using this term) of specific projections for memory formation or expression, and rather highlighted their importance in these processes, where appropriate.

9. Please discuss alternative interpretations of the data in the discussion.

Possible alternative interpretations of our data are twofold. First, we have tested whether the activity of the RSPag to PL projection at recent recall, which is not involved in memory retrieval itself, could have a role in reconsolidation (suggested by reviewer 2). Our new results (Figure 4 —figure supplement 3) suggests that this is unlikely. We now discuss these points in the revised version of the manuscript. Second, the degree of engram reactivation following our pathway-specific chemogenetic manipulations is, although significantly, only marginally impaired (as pointed out by reviewer 1), which implies that other regions or projections are also important in these processes and constrain the magnitude of the observed effects. We have now discussed this point in the revised version of the manuscript.